# ROYAL SOCIETY
# OPEN SCIENCE

health and disease and epidemiology

COVID-19, quarantine, screening

**Author for correspondence:**
Julian Peto
e-mail: julian.peto@lshtm.ac.uk

# Weekly COVID-19 testing with household quarantine and contact tracing is feasible and would probably end the epidemic

Julian Peto[1], James Carpenter[1], George Davey Smith[2], Stephen Duffy[3], Richard Houlston[4], David J. Hunter[5], Klim McPherson[6], Neil Pearce[1], Paul Romer[7], Peter Sasieni[8] and Clare Turnbull[4]

[1]Faculty of Epidemiology and Population Health, London School of Hygiene and Tropical Medicine, London, UK
[2]Bristol Population Health Science Institute, Bristol University, Bristol, UK
[3]Wolfson Institute of Preventive Medicine, Queen Mary University of London, London, UK
[4]Division of Genetics and Epidemiology, Institute of Cancer Research, London, UK
[5]Nuffield Department of Population Health, Oxford University, Oxford, UK
[6]New College, Oxford University, Oxford, UK
[7]Stern School of Business, New York University, New York, USA
[8]School of Cancer and Pharmaceutical Sciences, King's College London, London, UK

JP, 0000-0002-1685-8912

The COVID-19 epidemic can probably be ended and normal life restored, perhaps quite quickly, by weekly SARS-CoV-2 RNA testing together with household quarantine and systematic contact tracing. Isolated outbreaks could then be contained by contact tracing, supplemented if necessary by temporary local reintroduction of population testing or lockdown. Leading public health experts have recommended that this should be tried in a demonstration project in which a medium-sized city introduces weekly testing and lifts lockdown completely. The idea was not considered by the groups whose predictions have guided UK policy, so we have examined the statistical case for such a study. The combination of regular testing with strict household quarantine, which was not analysed in their models, has remarkable power to reduce transmission to the community from other household members as well as providing earlier diagnosis and facilitating rapid contact tracing.

> **Box 1.** 10 million tests per day are needed for weekly testing of the UK population.
>
> This is feasible with single-step RT-LAMP on saliva samples, which requires minimal equipment and training. A facility with about 100 staff could probably do 50 000 tests per day. If so, a city of 350 000 people could be served by a single laboratory. No government has considered this option because it was assumed that this level of testing is not technically possible even in a developed country like the UK. The technology can be implemented even in low-resource rural settings. A large number of RT-LAMP tests can be done in under an hour in a pan of warm water using a thermometer to maintain the temperature at about 63°C. The colour change showing a positive result can be read by eye.

# 1. Feasibility of regular testing in the UK and worldwide

Weekly viral RNA testing of the whole population with strict quarantine for households when an infection is detected has been proposed as a practical way to end lockdown while controlling the epidemic [1,2]. This was not evaluated in the models of non-pharmaceutical interventions that informed UK Government policy because universal testing was thought to be unfeasible. One modelling team dismissed the idea [3] and did not include either regular testing or household quarantine in their model [4]. Another [5] modelled partially effective quarantine in 50% of households beginning when symptoms appear. Isothermal single-step RT-LAMP [6] (reverse transcriptase loop amplification) is an economical high-throughput alternative that is at least three times faster than conventional RT-PCR (reverse transcriptase polymerase chain reaction) for detecting SARS-CoV-2 RNA and does not require expensive equipment or expert staff. This could be implemented worldwide with saliva samples as soon as an adequate supply of reagents can be manufactured (box 1). A saliva sample is more practical for self-sampling than a nasal/throat swab and provides comparable [7] and possibly superior [8] sensitivity. Implementation details for weekly sample collection and quarantine arrangements in a demonstration study and in subsequent national roll-out were outlined in an open letter to the UK Government on 10 April [2], with an estimated cost over a few months at about £1 billion per month for weekly testing of the whole UK population. This is a small fraction of the long-term economic costs of a continuing epidemic. Subsequent evidence that RT-LAMP testing on saliva self-samples is a viable (and cheaper) alternative to RT-PCR on nasal/throat swabs meets the two main logistical objections to that original proposal.

# 2. Effect on R of household quarantine based on weekly testing

We adopt the quantitative assumptions underpinning simulations of the situation prior to lockdown [5,9]. Further data on the unmitigated epidemic can no longer be collected, as all countries have now adopted control measures of varying stringency. In addition to weekly testing we also assume that an immediate test will be available on request for those developing COVID-19 symptoms. In the UK, about one-third of transmissions are thought to occur in households [5], so ignoring within-household heterogeneity in external contact rates the estimated reproduction number of 2.4 [5] before lockdown would fall to about 1.6 in the absence of household transmission. The mean serial interval is about 6.5 days [9], so quarantining when the first case in a household is detected will prevent the great majority of transmissions to the community from other household members infected by the case because they will not have been infectious for long, if at all. With 100% testing and quarantine compliance and 100% test sensitivity the proportion of community transmissions prevented by weekly testing would therefore have to be slightly greater than 38% (0.6/1.6) to bring the reproduction number R below 1 and control the epidemic. If the reproduction number is 3, removal of one-third due to household transmissions would reduce it to about 2, so more than 50% of community transmissions must also be prevented to bring R below 1. The simulated effect of self-isolation an average of 1.2 days after developing symptoms was a 51% reduction in transmissions [9]. Together with weekly testing to detect presymptomatic and asymptomatic infections, the majority of community transmissions may thus be prevented. However, this large predicted reduction in R depends on strong assumptions about the duration and level of infectiousness before and after developing symptoms [9]. The 95% confidence interval of the estimated proportion of transmissions that were from presymptomatic or asymptomatic cases was from 25 to 69% in a modelling study in China [10], and estimates from other studies span a similar range [11]. The median duration of viral shedding was 20

> **Box 2.** Ending lockdown in a whole-city demonstration study.
>
> About one-third of transmissions are within households. This large contribution to epidemic growth would be almost eliminated by strict household quarantine. Self-quarantine as soon as COVID-19 symptoms are noticed will further reduce transmissions. Contact tracing and mobile phone apps can have a large additional effect and would be even more effective within a population whose weekly test results are already available online. Whether the combined effect would control the epidemic can only be determined by a demonstration study in which a whole city is tested weekly and ends lockdown.

days in Wuhan inpatients [12] but a model fitted to Italian data assumed a much shorter duration in the community [13]. The distribution over all community transmissions of the time from the case becoming infectious to transmission determines the proportion of community transmissions that can be prevented by introducing weekly testing. In the absence of regular testing this poorly characterized prior distribution also depends on current compliance with self-quarantine when symptoms develop and the proportion of asymptomatic contacts already being identified and isolated. The effect on the reproduction number R of strict household quarantine based only on weekly testing without contact tracing is summarized by the equations in the statistical footnote.

## 3. Contact tracing, compliance and overall reduction in R

The large additional effect of contact tracing on transmission [11] would be further enhanced at lower cost within a population with weekly test results already available online. Mobile phone apps will further reduce transmissions. Self-quarantine as soon as symptoms appear is predicted to have a large effect [9], although evidence on whether infectiousness increases sharply when symptoms are developing seems contradictory [13,14]. The reproduction number is therefore likely although not certain to be reduced below 1 taking account of false negative tests and non-compliance. Whether a continuing epidemic in the minority who choose not to be tested would contribute substantially to infections among those who are regularly tested will depend on the extent to which they interact socially. Requiring evidence of a recent negative test for access to restaurants, bars and other public venues would reduce this hazard to those who are tested regularly, and would also encourage participation in testing.

## 4. Making strict household quarantine acceptable

RT-PCR and RT-LAMP both have high sensitivity [6], and false negative cases are likely to be less infectious. The proportion of potentially preventable transmissions actually prevented by weekly testing, and hence whether the epidemic can be controlled, will therefore be determined largely by compliance with weekly testing and quarantine. An important advantage of weekly testing, supplemented by a rapid test when a household member develops symptoms, is that other household members can take precautions immediately and hence reduce their risk. This can be supported in several ways, including a repeat test on the same day to identify false positive tests and avoid unnecessary quarantine, appropriately managed hotel accommodation for infected people who want it, PPE for all household members, and helplines for medical and social support and furlough arrangements. Quarantine must be both safe and tolerable to achieve high compliance.

## 5. A demonstration study of weekly testing should begin immediately

The hypothesis that the combination of weekly testing with an earlier test if symptoms appear, strict household quarantine, contact tracing and mobile phone apps would end the epidemic is thus plausible. Few households would be affected, as following lockdown less than 1 in 1000 of the population are now infectious [15]. The impact cannot be reliably predicted by further modelling, so a demonstration study in which a whole city introduces weekly testing together with these other measures and ends lockdown is needed [2] (box 2). Testing should be voluntary, and if those who agree to be tested are deemed to have consented to household quarantine if a household member

tests positive individual informed consent would not be needed. Within a few weeks, the epidemic may be reduced to occasional outbreaks, but if prevalence falls more slowly testing may have to continue for three or more months. R may decline initially then increase as an increasing proportion of people stop observing social distancing. The aim of the experiment is to see whether R still remains below 1. Whether the economy can be rescued and tens or hundreds of thousands of unnecessary deaths in the UK and millions worldwide can be prevented may depend on the outcome. Whatever the effect, the results (including prospective serology and genetics in a large randomly invited sample) would improve the reliability of natural history modelling, informing measures to control this epidemic and also to prevent any future pandemic of an even more lethal new virus.

### Statistical footnote

A new case whose last test was negative has a positive test. Sensitive testing is once every $D$ days, the delay between test and reported result is $d$ days and in the absence of testing the probability distribution of time $s$ from becoming infectious to a community transmission is $S(s)$. Then

$$P = \text{Proportion of community transmissions after positive result} = \frac{1}{D}\sum_{t=0}^{D-1}\int_{s=D+d-t}^{\infty} S(s).\mathrm{d}s$$

and

$$R(\text{untested}) = R(\text{community}) + R(\text{household}).$$

With strict household quarantine $R(\text{household})$ is small, so

$$R(\text{tested}) \approx (1 - P.Q.C).R(\text{community}),$$

where $Q$ is test sensitivity and $C$ is compliance with testing.

Authors' contributions. J.P. wrote the first draft. All authors collaborated in revising and approving the final text.
Conflicts of interest. All authors declare no conflict of interest.
Funding. No funding was received for this work.

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
