## [Reviewer comments · Royal Society Open Science]

Review History

RSOS-200915.R0 (Original submission)

Review form: Reviewer 1

Is the manuscript scientifically sound in its present form?

Yes

Are the interpretations and conclusions justified by the results?

Yes

Is the language acceptable?

Yes

Do you have any ethical concerns with this paper?

No

Have you any concerns about statistical analyses in this paper?

No

Recommendation?

Accept with minor revision (please list in comments)

Comments to the Author(s)

This is a nicely written manuscript about preventing covid-19 through household quarantine. The idea is sound and numerically the decline of R from 2.4 to 1.6 seems justified by a reference #5 (in all honesty, I cannot find that information in #5 and I am going to believe the authors that it is there) .

The manuscript would be much stronger if the authors themselves have a simulation/model and the showed the actual results of that simulation. The mathematician in me is worried that the main idea of household quarantine, however sound and logical, may have a different impact than the authors claim. If the authors have some sort of simulation supporting their claims, it would improve the manuscript.

At the same time, I agree with the authors that "the impact cannot be reliably predicted by further modelling" and that the demonstration study would be much more useful than a computer simulation. So I do not have any strong objections against the paper accepted as is.

Review form: Reviewer 2 (Marion Rowland)**Is the manuscript scientifically sound in its present form?**

No

Are the interpretations and conclusions justified by the results?

No

Is the language acceptable?

Yes

Do you have any ethical concerns with this paper?

Yes

Have you any concerns about statistical analyses in this paper?

Yes

Recommendation?

Reject

Comments to the Author(s)

This is a very interesting concept but some clarity is needed around the following points

1. Is it proposed to model weekly testing based on a number of scenarios, or is it proposed to carry out a pilot study in a defined population and to carry out weekly tests using RT-LAMP.
2. What is the duration of the study- number of weekly cycles of testing?
3. How will repeat cycles of testing impact on your detection of positive cases
4. What is the diagnostic accuracy of the RT-LAMP in particular what is the false positive rate.
5. How does prevalence of COVID impact on diagnostic accuracy, particularly in asymptomatic carriers.
6. Is the current prevalence of COVID 19 high enough to confer acceptable diagnostic accuracy on RT-LAMP.
7. What is the precision of RT-LAMP and against what Gold Standard was it measured.
8. While the authors believe that 1 person can carry out 500 tests per day it would be important to provide the evidence for this including specimen processing time, length of time specimen

required to be in a water bath, number of tests that can be conducted per water bath, time required for quality assurance.

9. Regardless of the urgency or public interest of such a study suggesting 100% compliance or 100% testing in any study is very unrealistic. On balance getting 90% of eligible persons with an analysable test might be possible in a much smaller community but not in such a large community.

10. Had the authors factored in any of the practical costs including obtaining informed consent, and the practical costs research staff in conducting such a large study.

Review form: Reviewer 3

Is the manuscript scientifically sound in its present form?

No

Are the interpretations and conclusions justified by the results?

No

Is the language acceptable?

Yes

Do you have any ethical concerns with this paper?

No

Have you any concerns about statistical analyses in this paper?

No

Recommendation?

Reject

Comments to the Author(s)

In this manuscript, the authors suggested that the combination of weekly SARS-CoV-2 RNA testing together with household quarantine and systematic contact tracing is sufficient to control the COVID-19 epidemic in a city, without the need of lockdown. I don't think that their proposed method is feasible. I have the following comments:

- (1) In "statistical footnote", it's not clear why specificity and false-positive were excluded in the analysis. The formula of P is quite messy. What's the meaning of S(s)?
- (2) The author assumed 100% testing and quarantine compliance, which are not likely to happen in reality.
- (3) Effective reproduction number < 1 does not mean that the epidemic will be stopped, but just indicate the effectiveness of control measures. Even if reproduction number < 1 , local transmission can still occur.
- (4) The authors implicitly assumed that the testing capacities could be uniformly distributed to all population in a city. However, due to the heterogeneity in population structure, this uniform distribution looks infeasible.

Review form: Reviewer 4 (Arindam Basu)

Is the manuscript scientifically sound in its present form?

Yes

Are the interpretations and conclusions justified by the results?

No

Is the language acceptable?

Yes

Do you have any ethical concerns with this paper?

No

Have you any concerns about statistical analyses in this paper?

Yes

Recommendation?

Accept with minor revision (please list in comments)

Comments to the Author(s)

Your solution is reasonable and likely to benefit; besides being pragmatic. Several countries have 'in fact' used similar approaches. These cases need to be highlighted. However, your 'formula' is simplistic, and there is no justification for $t = D - 1$ as a realistic reduction in the proportion might be expected roughly in 5 days time so something like $t = D - 3$ might be more realistic given the disease dynamics.

Please consider revising to add results of simulations based on your formula.

Decision letter (RSOS-200915.R0)

05-Jun-2020

Dear Dr Peto,

The editors assigned to your paper ("Weekly Covid-19 testing with household quarantine and contact tracing is feasible and would probably end the epidemic") have now received comments from reviewers. We would like you to revise your paper in accordance with the referee and Associate Editor suggestions which can be found below (not including confidential reports to the Editor). Please note this decision does not guarantee eventual acceptance.

Three reviewers have provided commentary on the report, and we're grateful to them for doing so in such a rapid manner. Under normal (ie pre-COVID) circumstances, the editors would have to consider rejecting the paper outright, but given the timely nature of the research and proposal, we would like you to instead tackle the concerns raised - both in the manuscript itself, and also provide a point-by-point rebuttal to the referees' commentary. We also recognise that the manuscript is more of an opinion piece than a research paper. However, please do work to address the concerns raised by the reviewers -- while the ultimate decision to accept/reject will be on the scientific merits of the paper and will be made by the Editors, this decision is informed by the view of the referees.

Please submit a copy of your revised paper before 28-Jun-2020. Please note that the revision deadline will expire at 00.00am on this date. If we do not hear from you within this time then it will be assumed that the paper has been withdrawn. In exceptional circumstances, extensions may be possible if agreed with the Editorial Office in advance. We do not allow multiple rounds of revision so we urge you to make every effort to fully address all of the comments at this stage.

If deemed necessary by the Editors, your manuscript will be sent back to one or more of the original reviewers for assessment. If the original reviewers are not available, we may invite new reviewers.

- Data accessibility

If you wish to submit your supporting data or code to Dryad (<http://datadryad.org/>), or modify your current submission to dryad, please use the following link:
<http://datadryad.org/submit?journalID=RSOS&manu=RSOS-200915>

- Competing interests

- Authors' contributions

- Acknowledgements

- Funding statement

on behalf of Prof Pete Smith (Subject Editor)
openscience@royalsociety.org

Comments to Author:

Reviewers' Comments to Author:

Reviewer: 1

Comments to the Author(s)

This is a nicely written manuscript about preventing covid-19 through household quarantine. The idea is sound and numerically the decline of R from 2.4 to 1.6 seems justified by a reference #5 (in all honesty, I cannot find that information in #5 and I am going to believe the authors that it is there) .

The manuscript would be much stronger if the authors themselves have a simulation/model and the showed the actual results of that simulation. The mathematician in me is worried that the main idea of household quarantine, however sound and logical, may have a different impact than the authors claim. If the authors have some sort of simulation supporting their claims, it would improve the manuscript.

At the same time, I agree with the authors that "the impact cannot be reliably predicted by further modelling" and that the demonstration study would be much more useful than a computer simulation. So I do not have any strong objections against the paper accepted as is.

Reviewer: 2

Comments to the Author(s)

This is a very interesting concept but some clarity is needed around the following points

1. Is it proposed to model weekly testing based on a number of scenarios, or is it proposed to carry out a pilot study in a defined population and to carry out weekly tests using RT-LAMP.
2. What is the duration of the study- number of weekly cycles of testing?
3. How will repeat cycles of testing impact on your detection of positive cases

4. What is the diagnostic accuracy of the RT-LAMP in particular what is the false positive rate.
5. How does prevalence of COVID impact on diagnostic accuracy, particularly in asymptomatic carriers.
6. Is the current prevalence of COVID 19 high enough to confer acceptable diagnostic accuracy on RT-LAMP.
7. What is the precision of RT-LAMP and against what Gold Standard was it measured.
8. While the authors believe that 1 person can carry out 500 tests per day it would be important to provide the evidence for this including specimen processing time, length of time specimen required to be in a water bath, number of tests that can be conducted per water bath, time required for quality assurance.
9. Regardless of the urgency or public interest of such a study suggesting 100% compliance or 100% testing in any study is very unrealistic. On balance getting 90% of eligible persons with an analysable test might be possible in a much smaller community but not in such a large community.
10. Had the authors factored in any of the practical costs including obtaining informed consent, and the practical costs research staff in conducting such a large study.

Reviewer: 3

Comments to the Author(s)

In this manuscript, the authors suggested that the combination of weekly SARS-CoV-2 RNA testing together with household quarantine and systematic contact tracing is sufficient to control the COVID-19 epidemic in a city, without the need of lockdown. I don't think that their proposed method is feasible. I have the following comments:

- (1) In "statistical footnote", it's not clear why specificity and false-positive were excluded in the analysis. The formula of P is quite messy. What's the meaning of S(s)?
- (2) The author assumed 100% testing and quarantine compliance, which are not likely to happen in reality.
- (3) Effective reproduction number < 1 does not mean that the epidemic will be stopped, but just indicate the effectiveness of control measures. Even if reproduction number < 1 , local transmission can still occur.
- (4) The authors implicitly assumed that the testing capacities could be uniformly distributed to all population in a city. However, due to the heterogeneity in population structure, this uniform distribution looks infeasible.

Reviewer: 4

Comments to the Author(s)

Your solution is reasonable and likely to benefit; besides being pragmatic. Several countries have 'in fact' used similar approaches. These cases need to be highlighted. However, your 'formula' is simplistic, and there is no justification for $t = D - 1$ as a realistic reduction in the proportion might be expected roughly in 5 days time so something like $t = D - 3$ might be more realistic given the disease dynamics.

Please consider revising to add results of simulations based on your formula.

Author's Response to Decision Letter for (RSOS-200915.R0)

See Appendix A.

Decision letter (RSOS-200915.R1)

11-Jun-2020

Dear Dr Peto,

It is a pleasure to accept your manuscript entitled "Weekly Covid-19 testing with household quarantine and contact tracing is feasible and would probably end the epidemic" in its current form for publication in Royal Society Open Science.

Appendix A

Response to reviewers' comments. Statements in bold are points that have been incorporated in the revised paper. One new reference (#9) has been added (see response 2 to reviewer 1).

Reviewers' Comments to Author:

Reviewer: 1

Comments to the Author(s)

This is a nicely written manuscript about preventing covid-19 through household quarantine. The idea is sound and numerically the decline of R from 2.4 to 1.6 seems justified by a reference #5 (in all honesty, I cannot find that information in #5 and I am going to believe the authors that it is there).

Author response 1: Second para of Methods in ref 5 states: "With the parameterisation above, approximately one third of transmission occurs in the household, one third in schools and workplaces and the remaining third in the community. These contact patterns reproduce those reported in social mixing surveys (8)."

The manuscript would be much stronger if the authors themselves have a simulation/model and the showed the actual results of that simulation. The mathematician in me is worried that the main idea of household quarantine, however sound and logical, may have a different impact than the authors claim. If the authors have some sort of simulation supporting their claims, it would improve the manuscript.

Author response 2: Our case for a demonstration project is that weekly testing with immediate strict household quarantine at a positive test (plus immediate testing on request when symptoms appear) is likely to reduce R to about 1 and perhaps less. This is based on our modelling, which is summarised below. The substantial additional effect of contact tracing conducted in a population already registered by household with weekly test results available online is therefore likely to reduce R to well below 1, but neither the effects of regular testing nor the additional impact of contact tracing can be predicted reliably by further simulation. We do not believe, therefore, that the case for a demonstration project would be decisively strengthened or weakened by computer simulation, and this reviewer appears on reflection to agree (see reviewer's concluding comment below). Simulation would require dubious assumptions about various unknown parameters, notably the poorly characterised probability distribution $S(s)$ of the time s from becoming infectious to transmission in the real-world situation before lockdown ($S(s)$ is defined in the statistical footnote).

Modelling the effect of weekly screening and household quarantine excluding the effect of contact tracing

Our argument that when lockdown is lifted R is likely to be 1 or less in the absence of contact tracing is as follows. **We adopt the quantitative assumptions underpinning ref 5, which describes the Imperial College simulation of the situation prior to lockdown that guided Government policy. We have added the statement that further data on the unmitigated epidemic can no longer be collected, as all countries have now adopted control measures of varying stringency.** Ref 5 states: "Based on fits to the early growth-rate of the epidemic in Wuhan (10,11), we make a baseline assumption that $R_0=2.4$ but examine values between 2.0 and 2.6." We therefore assume that $R=2.4$ (we also modelled $r=3$), and that one third of transmissions are within households (see response 1 above). Transmissions are either within households or in the outside community, so $R=R(\text{community}) + R(\text{household})$. This implies that $R(\text{community})=1.6$. If test results are returned within a day, which is feasible with the central testing arrangements we describe, weekly testing will identify the first case in a household within 1 to 8 days of becoming infectious. (The statistical footnote describes the general case in which testing is every D days and the reporting delay is d days.) To reduce $R(\text{community})$ from approximately 1.6 to 1 requires that slightly more than 38% of community transmissions are prevented by weekly testing. **A subsequent**

simulation (Kucharski et al – new ref #9) estimated that rapid self-isolation when symptoms develop will produce a 51% reduction in community transmission. In a population with weekly testing with an earlier test immediately available on request if symptoms appear the reduction in community transmission is thus likely to be substantially greater than 51%, confirming that R is likely to be reduced to 1 or less even without the large additional effect of contact tracing. We also cite the estimated 6.5 day mean serial interval from this new ref #9. This reinforces our claim that strict household quarantine within a maximum of 8 days after becoming infectious will prevent the great majority of community transmissions from other household members infected by the first case in a household.

At the same time, I agree with the authors that "the impact cannot be reliably predicted by further modelling" and that the demonstration study would be much more useful than a computer simulation. So I do not have any strong objections against the paper accepted as is.

Author response 3: That is why we have not added a computer simulation, which would inevitably entail questionable assumptions and hence would obscure the straightforward conclusion that a demonstration study is needed. See also responses 2 and 3 to reviewer 4.

Reviewer: 2

Comments to the Author(s)

This is a very interesting concept but some clarity is needed around the following points

1. Is it proposed to model weekly testing based on a number of scenarios, or is it proposed to carry out a pilot study in a defined population and to carry out weekly tests using RT-LAMP.

Author response 1: The latter. We thought this was clear.

2. What is the duration of the study- number of weekly cycles of testing?

Author response 2: This will be an experiment to eliminate a new disease that is still poorly characterised with a novel intervention that has never been tried for any infection. We have added the statement that the epidemic may be reduced to occasional outbreaks within a few weeks, but if prevalence falls more slowly testing may have to continue for three or more months.

3. How will repeat cycles of testing impact on your detection of positive cases

Author response 3: Some infected cases missed either because the sample was inadequate or because they are newly infected and are not yet very infectious will be detected the following week, further reducing R. We expect R to fall well below 1 immediately, with a steady decrease in the number of new cases. R may then increase as an increasing proportion of people stop observing social distancing. The aim of the experiment is to see whether it still remains below 1.

4. What is the diagnostic accuracy of the RT-LAMP in particular what is the false positive rate.

Author response 4: Ref 6 reports perfect agreement between the SARS-CoV-2 RT-LAMP assay and RT-qPCR and shows that RT-LAMP has high sensitivity (118.6 copies of SARS-CoV-2 RNA per 25 µL reaction). The sensitivity of RT-LAMP is now being improved in several labs. False positives are rare, and as we mentioned could be virtually eliminated by an immediate repeat test.

5. How does prevalence of COVID impact on diagnostic accuracy, particularly in asymptomatic carriers.

Author response 5: Sensitivity, and hence the impact on R, are independent of disease prevalence. As mentioned, asymptomatic cases that are missed because they are shedding at a low level are likely to be less infectious, reducing the importance of false negatives.

6. Is the current prevalence of COVID 19 high enough to confer acceptable diagnostic accuracy on RT-LAMP.

Author response 6: As sensitivity and specificity are independent of disease prevalence we are not clear what the reviewer means by diagnostic accuracy. The impact on R is unaffected by false positives, although as stated these would be eliminated by immediate repeat testing to avoid unnecessary quarantine.

7. What is the precision of RT-LAMP and against what Gold Standard was it measured.

Author response 7: See responses 4-6. The comparison in ref 6 was against RT-qPCR.

8. While the authors believe that 1 person can carry out 500 tests per day it would be important to provide the evidence for this including specimen processing time, length of time specimen required to be in a water bath, number of tests that can be conducted per water bath, time required for quality assurance.

Author response 8. The precise staffing level required can only be determined in a demonstration study, as a facility for high throughput RT-LAMP has not yet been established. Most of the work would be specimen reception and handling, as the RT-LAMP reaction time for a batch of 96 samples (including positive and negative controls) is only about half an hour, plus a few minutes for reading and data entry. Current UK COVID-19 prevalence is less than 1 in 500, so the majority of batches will all be negative.

9. Regardless of the urgency or public interest of such a study suggesting 100% compliance or 100% testing in any study is very unrealistic. On balance getting 90% of eligible persons with an analysable test might be possible in a much smaller community but not in such a large community.

Author response 9: We did not assume 100% compliance and sample adequacy. We showed that R might be so low with 100% compliance and 100% test sensitivity that the epidemic would probably be ended quite rapidly allowing for these factors, which are modelled explicitly in the statistical footnote but cannot be predicted reliably.

10. Had the authors factored in any of the practical costs including obtaining informed consent, and the practical costs research staff in conducting such a large study.

Author response 10: The cost of RT-LAMP reagents is only about £1 per test, so most of the cost will be salaries and facilities. **The approximate overall cost was estimated at ~£1 billion per month for the whole UK in an earlier letter to the Government based on RT-PCR performed by a collaboration of private and academic labs (ref 2 is an abbreviated version which includes the relevant online citation). The cost is likely to be less with centralised RT-LAMP testing. The cost over a few months would thus be negligible compared with the long-term economic costs of the continuing epidemic.** This ignores the potential prevention of a hundred thousand or more eventual deaths in the UK alone. The aim of our paper is to argue that such a study should be conducted, not to present a detailed costed protocol. **We have added the proposal that testing should be voluntary, but those who agree to be tested would be deemed to have consented to household quarantine if a household member tests positive. Individual informed consent is therefore not required.**

Reviewer: 3

Comments to the Author(s)

In this manuscript, the authors suggested that the combination of weekly SARS-CoV-2 RNA testing together with household quarantine and systematic contact tracing is sufficient to control the COVID-19

epidemic in a city, without the need of lockdown. I don't think that their proposed method is feasible. I have the following comments:

(1) In "statistical footnote", it's not clear why specificity and false-positive were excluded in the analysis. The formula of P is quite messy. What's the meaning of S(s)?

Author response 1: False positives are not modelled because they have no effect on epidemic control. As stated any false positive results would in any case be eliminated by immediate repeat testing. Their only effect is thus a day of unnecessary household quarantine while the test is repeated. The formula for P cannot be simplified – what does "messy" mean? As stated, S(s) is the probability distribution over all community transmissions of the time s from becoming infectious (i.e. shedding detectable virus) to a community transmission in the absence of the proposed intervention.

(2) The author assumed 100% testing and quarantine compliance, which are not likely to happen in reality.

Author response 2: We did not assume 100% compliance. We showed that R would be likely to be so low with 100% compliance and 100% test sensitivity that the epidemic would probably be ended quite rapidly allowing for compliance and test sensitivity, which are modelled explicitly in the statistical footnote.

(3) Effective reproduction number < 1 does not mean that the epidemic will be stopped, but just indicate the effectiveness of control measures. Even if re production number < 1 , local transmission can still occur.

Author response 3: $R < 1$ implies that incidence and hence prevalence will fall steadily, eventually reaching a level at which intensive contact tracing, supplemented if necessary by temporary local reintroduction of population testing or lockdown, can eradicate isolated outbreaks.

(4) The authors implicitly assumed that the testing capacities could be uniformly distributed to all population in a city. However, due to the heterogeneity in population structure, this uniform distribution looks infeasible.

Author response 4: The population will be divided into the majority who are tested regularly and those who choose not to be tested. We have added a paragraph on transmissions from the untested minority to the regularly tested majority, and the suggestion that this would be reduced (and compliance with testing would be increased) if evidence of a recent negative test were required for admission to supermarkets, restaurants and other public venues.

Reviewer: 4

Comments to the Author(s)

Your solution is reasonable and likely to benefit; besides being pragmatic. Several countries have 'in fact' used similar approaches. These cases need to be highlighted.

Author response 1: No large country has tried to eliminate this or any other established epidemic by repeated testing, household quarantine and contact tracing of the whole population while lifting all other control measures.

However, your 'formula' is simplistic, and there is no justification for $t = D - 1$ as a realistic reduction in the proportion might be expected roughly in 5 days time so something like $t = D - 3$ might be more realistic given the disease dynamics.

Author response 2: This is a misunderstanding. The summation in the equation for P is from $t=0$ to $t=D-1$ because there are D days between tests ($D=7$ for weekly testing). The integral is the proportion of

transmissions that could be prevented if infectiousness started t days after the preceding negative test (allowing for the fixed reporting delay of d days), so the sum is multiplied by the probability $1/D$ that the case became infectious t days after their last test ($t=0,1,2,\dots,D-1$). All this is independent of disease dynamics, which (together with behaviour) determines the probability distribution over all community transmissions $S(s)$ of the time s from becoming detectable to transmission. As we state the formula for P does not provide a basis for reliable simulation because $S(s)$ cannot be estimated reliably. That is why we recommend an immediate demonstration study rather than further modelling.

Please consider revising to add results of simulations based on your formula.

Author response 3: The main conclusions of our paper are that the combination of weekly testing, household quarantine and contact tracing is likely but not certain to end the epidemic, and that further simulation cannot predict the effect reliably. Even if the demonstration study shows that the epidemic cannot be ended without social distancing measures the additional effect of a household based population register with weekly testing results on the effectiveness of each other intervention (earlier isolation, earlier household quarantine, and more efficient contact tracing) will be substantial, and the unbiased population-based results will provide the basis for greatly improved simulation and prediction.